# Micro-Doppler Effect and Sparse Representation Analysis of Underwater Targets

**DOI:** 10.3390/s23198066

**Published:** 2023-09-25

**Authors:** Yan Lu, Siwei Kou, Xiaopeng Wang

**Affiliations:** 1School of Electronic and Information Engineering, Lanzhou Jiaotong University, Lanzhou 730070, China; tyccly123@126.com (Y.L.); wangxiaopeng@mail.lzjtu.cn (X.W.); 2School of Marine Science and Technology, Northwestern Polytechnical University, Xi’an 710072, China

**Keywords:** micro-Doppler effect, sparse representation, time–frequency atom, time–frequency distribution, underwater target recognition

## Abstract

At present, the micro-Doppler effects of underwater targets is a challenging new research problem. This paper studies the micro-Doppler effect of underwater targets, analyzes the moving characteristics of underwater micro-motion components, establishes echo models of harmonic vibration points and plane and rotating propellers, and reveals the complex modulation laws of the micro-Doppler effect. In addition, since an echo is a multi-component signal superposed by multiple modulated signals, this paper provides a sparse reconstruction method combined with time–frequency distributions and realizes signal separation and time–frequency analysis. A MicroDopplerlet time–frequency atomic dictionary, matching the complex modulated form of echoes, is designed, which effectively realizes the concise representation of echoes and a micro-Doppler effect analysis. Meanwhile, the needed micro-motion parameter information for underwater signal detection and recognition is extracted.

## 1. Instruction

Victor C. Chen introduced the concept of micro-Doppler effects in radar [1]. The micro-Doppler effect is a physical phenomenon caused by micro-motions, such as vibrations and rotations of target components, which are inherent features of moving targets. Extracting micro-Doppler features is of great significance for improving the performance of target detection and recognition. At present, there are many studies on this topic in the radar and air sonar fields [2,3,4]. However, in the field of underwater acoustic signal processing, it is only in the beginning stages [5,6]. When a large underwater target, such as a submarine, unmanned underwater vehicle, or submersible, is underway, its components will vibrate, rotate, and perform other complex periodic micro-motions due to the action of forces from the engine, propeller, and other mechanical equipment, resulting in a micro-Doppler effect. The micro-Doppler effects of underwater targets are more complicated, due to the influence of an inhomogeneous water medium and turbulence on the vibration of the target shell and the rotation of the propeller.

The magnitude of micro-Doppler frequency shifts is directly proportional to the product of the vibration rate and displacement offset and inversely proportional to the emission signal wavelength [1]. The Doppler modulation of vibration can be observed if the product is large enough and the signal wavelength is small enough.

Micro-motions produce complex modulation of sonar emission signals, and the target echo is a nonstationary time-varying signal. The main theories of nonstationary signal analysis are joint time–frequency distribution [7] and instantaneous frequency analysis [8]. Different kernel functions can be used to analyze the local joint time–frequency information of multi-component signals successfully, but the inherent cross-term problem of quadratic time–frequency distribution has not been solved. Instantaneous frequency analysis can obtain micro-Doppler frequency estimation, but the analytical signal obtained by the Hilbert transform is only suitable for analyzing single-component signals. For multi-component signals superposed by the reflected waves from target components, determining how to analyze and interpret the modulation components contained in the echo and extracting the micro-Doppler features is still a challenge.

In recent years, the sparse representation theory of signals has attracted broad attention [9,10], and has achieved good application effects in many aspects of signal processing and deep learning [11,12,13,14]. Mallat and Zhang proposed a new time–frequency distribution method based on the Wigner Ville distribution [15]. The method constructed a time–frequency distribution without interatomic cross-term interference and with good time–frequency aggregation. This time–frequency distribution can adaptively adjust the characteristic parameters of time–frequency atoms with the local characteristics of the analyzed signal. Therefore, the time–frequency distribution characteristics of multi-component time-varying signals can be revealed, which provides a method for analyzing micro-Doppler effects.

In this paper, the micro-Doppler effects of underwater targets are studied. According to the micro-motion analysis results of underwater targets, the echo models of some typical micro-motion components are established. A time–frequency atomic dictionary, the MicroDopplerlet dictionary, is designed. The modulation law and micro-Doppler effects of echoes are analyzed by sparse reconstruction combined with time–frequency distributions.

## 2. Micro-Motion Analysis of Underwater Targets

The vibrating shell and rotating propeller are the main generators of the micro-motions of underwater targets. The magnitude of the micro-motion frequency and displacement are the basis of micro-Doppler effects. The propeller blade is long and has a very high tip velocity, causing a large amount of velocity and displacement [16]. Table 1 calculates the micro-Doppler frequency shift of 5-blade and 7-blade propellers and shells under typical conditions. The results show that propellers have obvious micro-Doppler effects.

The shell of underwater targets can be classified as a single- or double-layer cylindrical shell. Aron correctly described the vibration problem of cylindrical shells for the first time [17]. Flugge [18] and Love [19] proposed methods to calculate the vibration of a thin shell with an infinite length and fixed length, establishing the famous Love first approximation theory and obtaining the calculation formula for the free vibration frequency of an infinite-length cylindrical shell. Subsequently, based on the Love theory, many scholars have developed various theories for approximate cylindrical shells [20,21,22]. The results show that low frequencies correspond to the large radial displacement, middle frequencies correspond to the axial displacement component, and high frequencies correspond to the circumferential displacement. Yamada [23] used the transfer matrix method to calculate the free vibration of a double-layer cylindrical shell. Raheb and Wagner [24] established the mechanical equilibrium equation of a double-layer stiffened shell. The force of a stiffened double-layer cylindrical shell is expressed in the form of shell displacement, which provides a theory and method for the calculation of shell displacement.

On the other hand, Tang et al. conducted theoretical research on the vibration characteristics of ring-ribbed cylindrical shells under different boundary conditions and discussed the influence of different parameter changes on the natural frequency of shell structures [25]. Shang et al. used ANSYS software to obtain the normal displacement, normal vibration velocity, surface sound pressure, normal sound intensity, and other parameters of the outer surface of double-layer cylindrical shells [26]. The submarine is usually simplified to the model shown in Figure 1. Wang [27] carried out experiments on the vibration characteristics of different parts of a simplified model of a single-layer shell. When a 500 N external force was applied to the shell, the vibration displacement of the hemisphere shell A was 3.98 microns, the vibration displacement of the middle cylindrical shell B was 8.91 microns, and the vibration displacement of the conical shell C was 12.59 microns. Huang [28] conducted a numerical simulation experiment on the vibration characteristics of double-layer shells. When a 1000 N external force was applied to the shell, the vibration displacement of the hemisphere shell A was 5.62 microns, the vibration displacement of the middle cylindrical shell B was 7.94 microns, and the vibration displacement of the conical shell C was 3.16 microns.

The propeller and shafting–shell coupled system have an important influence on the shell vibration. The unsteady external excitation force of the propeller during operation and the friction force of the shafting and bearing caused by the propeller rotation are important excitation sources for the vibration of the shell structure. At present, scholars have paid attention to the exciting force of the propeller on the shell [29,30,31,32,33]. Hassan et al. conducted an experimental test on the hydrodynamic performance of a 5-blade high-skew propeller [16]. The results showed that the propeller’s excitation force could reach 158,750 N when the advance coefficient was 0.3, and the rotational velocity was 600 r/m. This thrust force is amplified by shafting and will cause the shell vibration greatly. The research results of reference [34] showed that the unsteady excitation force of the propeller can be amplified at the natural vibration frequency, and the excitation force transmitted to the shell at the first natural frequency can be amplified nearly 20 times. Zhang [35] established the coupled shafting–hull structure model and obtained an unsteady excitation force from a 7-blade propeller with a rotating velocity of 110 r/m using a semi-empirical formula, among which the longitudinal unsteady excitation force was about 700 N.

Based on the above research results, if the propeller excitation force is known, the excitation force transmitted to the shell can be obtained through the shafting amplification characteristics. Then, according to the displacement generated by the unit excitation force, the vibration displacement of the shell can be calculated.

The estimation results of the micro-motion amplitude, micro-motion frequency, and micro-Doppler frequency of propellers and different components of shells are given in Table 1. When the wavelength is 50 mm, according to Equations (11) and (16), the calculated micro-Doppler frequency is above 1 kHz. It can be seen that the propeller has a high velocity, large displacement, and significant micro-motion characteristics, and the micro-Doppler frequency shift is large. The vibration amplitude of the submarine shell can reach a centimeter level, and the micro-Doppler frequency shift can be hundreds of hertz.

## 3. Micro-Doppler Effect and Echo Models of Typical Components

Generally speaking, micro-Doppler effects are not only related to the wavelength and the product of micro-motion frequency and displacement but also to the observation direction of sonar. When studying micro-Doppler effects, the included angle between the vibration direction and the incident wave direction should be considered to establish the echo model. However, for simple harmonic vibration points and planes and rotation propellers, the simplified models based on ignoring the included angle are still universal. Simplified models can describe the complex modulation law of micro-Doppler effects. The simplified echo models of these three types are established below.

Based on the comprehensive research results, the calculation formula for vibration amplitude is:(1)Δl=K×F×ρ,
where F is the applied propeller excitation force, ρ is the vibration displacement generated by a unit Newtonian force, and K is the amplification factor of the excitation force transmitted from the first natural frequency of the propeller to the shell.

### 3.1. Scattering Point of Simple Harmonic Vibration

Smaller target components can be regarded as scattering points. When a scattering point vibrates, the echo phase is modulated, and the frequency changes periodically. The sideband frequency near the carrier frequency is generated, the micro-Doppler effect. The geometric relationship between vibration point and sonar is established as shown in Figure 2. The target coordinate system $x^{\prime }y^{\prime }z^{\prime }$ is the translation of the sonar coordinate system xyz along the z axis.

Supposing that the scattering point $P_{t}$ vibrates along the z axis, and the equilibrium position is $\;O\;’(0,0,z_{0})$, $A_{v}$ is the amplitude, $f_{v}$ is the vibration frequency, and $\omega_{v}=2\pi f_{v}$ is the angular frequency. The simple harmonic vibration of scattering points can be regarded as a spring oscillator system, and the displacement law is obtained from kinematics:(2)$${z^{\prime }}_{t}=A_{v}\sin(\omega_{v}t+\phi ),$$
where ϕ is the initial phase. The displacement is a sine function varying with time t. The distance from the scattering point to the sonar is:(3)$$r_{p}(t)=r_{0}+{z^{\prime }}_{t}=z_{0}+A_{v}\sin(\omega_{v}t+\phi ),$$

Let $s(t)=\tilde{u}e^{j2\pi f_{0}t}$ be the sonar emission signal. Where $f_{0}$ is the carrier frequency and $\tilde{u}$ is the complex envelope. Under ideal conditions, the echo is the delay of the emission signal, that is:(4)$$x\left(t\right)=\sigma \tilde{u}e^{-j\frac{4\pi }{\lambda }z_{0}}e^{j[2\pi f_{0}t-\frac{4\pi }{\lambda }A_{m}\sin\left(\omega_{v}t+\varphi \right)]},$$
where σ is the reflection coefficients. $\Phi (t)=2\pi f_{0}t-\frac{4\pi }{\lambda }A_{m}\sin(\omega_{v}t+\varphi )$ is the phase function, and the time derivative of Φ(t) is instantaneous frequency, which is:(5)$$f(t)=\frac{1}{2\pi }\frac{d\Phi }{dt}=f_{0}-\frac{2}{\lambda }A_{m}\omega_{v}\cos(\omega_{v}t+\varphi ),$$

The phase modulation causes micro-Doppler effects. The micro-Doppler frequency is:(6)$$f_{md}=-\frac{2}{\lambda }A_{m}\omega_{v}\cos(\omega_{v}t+\varphi ),$$

### 3.2. Scattering Plane of Simple Harmonic Vibrations

The large area shell vibration can be regarded as a vibration scattering plane. The amplitude and phase of echoes are all modulated. The amplitude flash and the frequency change periodically. The geometrical relationship between the vibration plane and the observation sonar is shown in Figure 3. The center of the vibration plane is located at $\;O\;’(0,0,z_{0})$, perpendicular to the z axis. The scattering points on the plane vibrate along the z axis. The vibration angular frequency $\omega_{v}$ is the same, but the amplitude is different. The amplitude distribution function is $A_{v}(x^{\prime },y^{\prime })$. The echo model can be established by analyzing the displacement law of simple harmonic points.

At time t, the displacement of any scattering point $P_{t}(x^{\prime },y^{\prime },0)$ on the vibration plane away from the equilibrium position is $z_{t}{}^{\prime }=A_{v}(x^{\prime },y^{\prime })\sin\omega_{v}t$. The distance from the sonar to the scattering point is:(7)$$r_{p}(t)={\left[{x^{\prime }}^{2}+{y^{\prime }}^{2}+{\left(z_{0}+{z^{\prime }}_{t}\right)}^{2}\right]}^{1/2},$$

Substituting Equation (6) into ${z^{\prime }}_{t}$ in the case of $z_{0}>>{z^{\prime }}_{t}$, the above formula becomes approximately:(8)$$r_{p}(t)\approx z_{0}+A_{v}(x^{\prime },y^{\prime })\sin\omega_{v}t,$$

The reflected wave of $P_{t}$ can be expressed as:(9)$$x_{p}(t)=\sigma \tilde{u}e^{-j\frac{4\pi }{\lambda }z_{0}}e^{j[2\pi f_{0}t-\frac{4\pi }{\lambda }A_{v}(x^{\prime },y^{\prime })\sin(\omega_{v}t)]},$$

A double integral of the reflected wave is carried out over the whole plane. The echo is:(10)$$x(t)=\sigma \tilde{u}e^{j(2\pi f_{0}t-\frac{4\pi z_{0}}{\lambda })}\iint_{S}e^{-j\frac{4\pi }{\lambda }A_{v}(x^{\prime },y^{\prime })\sin(\omega_{v}t)}ds,$$
where S is the integral region and ds is the area element. For a circular plane of radius R, let the amplitude distribution function be $A_{v}(r^{\prime })=A_{v}\frac{R^{2}-{r^{\prime }}^{2}}{R^{2}}$. After integration, the echo is:(11)$$\begin{array}{c} x(t)=\sigma \tilde{u}\pi R^{2}e^{-j\frac{4\pi z_{0}}{\lambda }}e^{j[2\pi f_{0}t-\frac{2\pi }{\lambda }A_{v}\sin(\omega_{v}t)]}\\ \cdot \sin c[\frac{2\pi }{\lambda }A_{v}\sin(\omega_{v}t)] \end{array},$$

Due to the existence of the sinc function, the amplitude presents the characteristics of a time domain flash. The phase is modulated by a sinusoidal function. The micro-Doppler frequency is:(12)$$f_{md}=\frac{1}{2\pi }\frac{d}{dt}\Phi (t)-f_{0}=-\frac{1}{\lambda }A_{v}\omega_{v}\cos(\omega_{v}t),$$

### 3.3. Blades of Rotation Propeller

Similar to the vibration plane, when the propeller rotates, the blade echo is modulated in both amplitude and phase. The geometric relationship between blades and sonar is shown in Figure 4. Let the blade’s length be L, the center is located at $\;O\;’(0,0,z_{0})$, and the blades rotate at an angular velocity of $\omega_{v}$.

Let the distance from any scattering point $P_{t}$ to $O^{\prime }$ on the blade be $l_{p}$. If the position is $P_{0}(0,{y^{\prime }}_{0},{z^{\prime }}_{0})$ at t=0 and the position is $P_{t}(0,{y^{\prime }}_{t},{z^{\prime }}_{t})$ at t=t, then ${y^{\prime }}_{t}=-l_{p}\sin(\psi_{0}+\omega_{v}t)$ and ${z^{\prime }}_{t}=l_{p}\cos(\psi_{0}+\omega_{v}t)$, where $\psi_{0}$ is the initial rotation angle. In the sonar coordinate system, the coordinate of $P_{t}$ is $\left({y^{\prime }}_{t},z_{0}+{z^{\prime }}_{t}\right)$, that is, $P_{t}[-l_{p}\sin(\psi_{0}+\omega_{v}t),z_{0}+l_{p}\cos(\psi_{0}+\omega_{v}t)]$. The distance from $P_{t}$ to O is:(13)$$r_{p}(t)={\left[z_{0}^{2}+l_{p}^{2}+2z_{0}l_{p}\cos(\psi_{0}+\omega_{v}t)\right]}^{1/2},$$

Since $z_{0}>>l_{p}$, the distance is approximately:(14)$$r_{p}(t)\approx z_{0}+l_{p}\cos(\psi_{0}+\omega_{v}t),$$

The reflected wave of scattering point $P_{t}$ is:(15)$$x_{p}(t)=\sigma \tilde{u}e^{-j\frac{4\pi }{\lambda }z_{{}_{0}}}e^{j[2\pi f_{0}t-\frac{4\pi }{\lambda }l_{p}\cos(\psi_{0}+\omega_{v}t)]},$$

The reflected wave is integrated into the blade length L, and the echo of the blade is:(16)$$\begin{array}{c} x(t)=\sigma \tilde{u}Le^{-j\frac{4\pi }{\lambda }z_{0}}e^{j[2\pi f_{0}t-\frac{2\pi }{\lambda }L\cos(\psi_{0}+\omega_{v}t)]}\\ \cdot \sin c[\frac{2\pi }{\lambda }L\cos(\psi_{0}+\omega_{v}t)] \end{array},$$

The micro-Doppler frequency is:(17)$$f_{md}=-\frac{1}{\lambda }L\omega_{v}\sin(\psi_{0}+\omega_{v}t),$$

The size of the micro-Doppler frequency is related to the blade length, rotation angular velocity, and wavelength. The echo amplitude flash and micro-Doppler shift are unique characteristics of underwater targets in the time and frequency domains. Micro-Doppler features are considered important features of underwater target recognition. The flash characteristics can be observed in the time domain, while the micro-Doppler frequency shift requires joint time–frequency analysis.

For a propeller with N identical blades, N blades are equally spaced. The initial rotation angle of each blade is:(18)$$\psi_{k}=\psi_{0}+(k-1)\frac{2\pi }{N}\begin{array}{cc} , & k=1,2,\cdots ,N \end{array},$$

According to the superposition principle, the echo is:(19)$$\begin{array}{c} x(t)=\sum_{k=1}^{N}e^{-j\frac{2\pi }{\lambda }L\cos(\psi_{0}+(k-1)\frac{2\pi }{N}+\omega_{v}t)}\\ \cdot \sin c[\frac{2\pi }{\lambda }L\cos(\psi_{0}+(k-1)\frac{2\pi }{N}+\omega_{v}t)] \end{array},$$

The micro-Doppler frequency of the kth blade is:(20)$$f_{md}^{k}=-\frac{L}{\lambda }\omega_{v}\sin(\psi_{0}+(k-1)\frac{2\pi }{N}+\omega_{v}t),$$

## 4. Sparse Reconstruction Combined with Time–Frequency Distribution

The echo of a moving target is a nonstationary signal superimposed by many complex modulation components. It is difficult to use joint time–frequency distribution to analyze micro-Doppler effects and local joint time–frequency information. Sparse decomposition can better separate multi-component signals and effectively reveal the time-varying characteristics of nonstationary signals. It provides a method for analyzing micro-Doppler effects and for feature extraction of underwater targets.

This paper presents a method of sparse reconstruction combined with time–frequency distributions. A time–frequency atomic dictionary is designed according to the modulation form of micro-motion components’ echo. Multi-component signals are separated into single-component signals by sparse decomposition of the echo. The Wigner Ville distribution is used to analyze the separated single-component signals. This method constructs a time–frequency distribution without interatomic cross-term interference and with good time–frequency aggregation. It can analyze the micro-Doppler effects of echoes well.

### 4.1. Sparse Representation of Echo Data

Supposing that an existing overcomplete dictionary $D=\{g_{r}(t),r=1,2,...,K\}$, where $g_{k}(t)$ represent atoms. In the Hilbert space $H=R^{N}$ where K>N, for any echo data x(t)∈H, if M atoms are selected adaptively in dictionary D, the nonlinear approximation of the echo is:(21)$$x(t)=\sum_{r\in I_{M}}c_{r}g_{r}(t),$$
where the set of atoms corresponding to the integer subscript set $I_{M}$ is $B=\{g_{r}(t),r\in I_{M}\}$. Let $\Gamma =\{I_{m}\}$ be the set of all subscripts and $c=\{c_{r},r\in I_{M}\}$ be the nonlinear representation coefficients. If the potential of the subscript set is $Card(I_{M})<N$, then c is the sparse representation of the signal in dictionary D. In particular, if $Card(I_{M})=\underset{I_{m}\in \Gamma }{\min}Card(I_{m})$, c is called an optimal sparse representation. Equation (20) is written in matrix form x=Dc, where x={x(t),t=0,1,...,N−1}. The solution of coefficient c can be expressed as a non-convex optimization model:(22)$$\underset{c}{\min}{\left\|c\right\|}_{0},s.t.x=Dc,$$
where $||\cdot |{|}_{0}$ is the L0 norm. The sparse representation results of MP after M iterations are:(23)$$x(t)=\sum_{m=0}^{M-1}\langle R^{m}x(t),g_{r_{m}}\rangle g_{r_{m}}+R^{M}x(t),$$

In the formula, $c_{m}=\langle R^{m}x(t),g_{r_{m}}\rangle$ is the sparse decomposition coefficient, where 〈⋅〉 represents the inner product, $g_{r_{m}}$ is the optimal atom chosen in each iteration, $R^{M}x(t)$ is the residual signal obtained after the Mth iteration, and $R^{0}x(t)=x(t)$. Ignoring the residual signal, sparse representation of the echo is solved by the Wigner Ville distribution, namely:(24)$$\begin{array}{c} {\mathrm{WVD}}_{x}(t,\omega )=\sum_{m=0}^{M-1}{\left|c_{m}\right|}^{2}{\mathrm{WVD}}_{g_{r_{m}}}(t,\omega )\\ +\sum_{m=0}^{M-1}\sum_{n=0,n\neq m}^{M-1}c_{m}c_{n}^{*}{\mathrm{WVD}}_{g_{r_{m}}g_{r_{n}}}(t,\omega ) \end{array},$$

The first term, on the right of the above equation, is the sum of the Wigner Ville distribution of the self-term between atoms, and the second term is the cross-term between atoms. The first term can correctly represent the time–frequency distribution structure of signal x(t), and can be obtained using the optimal atom $g_{r_{m}}$ selected in each iteration and the decomposition coefficient $c_{m}$. Therefore, a time–frequency distribution without interatomic cross-term interference can be obtained by retaining the first term of the above equation:(25)$$E_{x}(t,\omega )=\sum_{m=0}^{M-1}{\left|c_{m}\right|}^{2}{\mathrm{WVD}}_{g_{r_{m}}}(t,\omega ),$$

Clearly, the properties of this time–frequency distribution are determined by the Wigner Ville distribution, and it has a good time–frequency aggregation and time–frequency resolution.

### 4.2. Design of Time–Frequency Atomic Dictionary

The dictionary design is an important aspect of sparse decomposition and the key to analyzing micro-Doppler effects. The design of time–frequency atoms should benefit both time–frequency representation of nonstationary signals and the concise representation of signals.

The micro-Doppler effect is a complex modulation form. The time–frequency characteristic of echoes is an oscillation curve in the time–frequency plane. Although Gabor, Chirplet, Dopplerlet, and other traditional analytical dictionaries have certain versatility, they cannot easily and succinctly represent such complex modulation signals. Based on the time–frequency characteristics of the signal, a kind of time–frequency atom can be designed to match the echo modulation form to represent the echo succinctly. To match the modulation form of the echo, ignoring the known carrier frequency, the time–frequency atoms can be constructed as follows:(26)$$g_{\gamma }=w(t-u)\exp\{-j\frac{4\pi }{\lambda }A\sin[\omega (t-u)]\},$$
where w is a window function to fit the envelope of the signal, u is a displacement factor, and A is the oscillation amplitude factor. $\omega =2\pi f_{r}$ is the oscillation frequency factor.

The amplitude and phase of an echo are all modulated. According to the modulation form of an echo, the constructed time–frequency atoms are:(27)$$\begin{array}{c} g_{\gamma }=w(t-u)\cdot \sin c\{\frac{2\pi }{\lambda }A\sin[\omega (t-u)]\}\\ \cdot \exp\{-j\frac{2\pi }{\lambda }A\sin[\omega (t-u)]\} \end{array},$$

The time–frequency atoms represented in Equations (25) and (26) are called MicroDopplerlet atoms. The time–frequency distribution of an echo can be obtained by substituting the optimal atoms selected by MP into Equation (24). The designed time–frequency atoms match the echo modulation form. Therefore, the atoms can describe the complex modulation law and time–frequency distribution of the echo, which can represent the echo succinctly. When the echo is sparsely decomposed, the instantaneous frequency of the selected optimal time–frequency atom equals the micro-Doppler frequency of each echo component. Therefore, the micro-Doppler frequency is:(28)$$f_{md}=\frac{1}{2\pi }\frac{d}{dt}\Phi (t)=\frac{C}{\lambda }A\omega \cos[\omega (t-u)],$$
where $\Phi (t)=\arg\{g_{r}\}$ and arg{⋅} means extracting the phase information. The value of C is determined by amplitude scintillation and amplitude flash. For a simple harmonic vibration point, C=2, and for a simple harmonic vibration plane and rotation propeller, C=1.

When the envelope and wavelength parameters of the emission signal are known, the time–frequency atoms have only three parameters: A, ω, and u. Discretizing these three parameters, the set obtained is the time–frequency atomic dictionary, the MicroDopplerlet dictionary. The discretizing method is given below.

Constrained by equal interval sampling, u is the equal interval value, that is, $u=n/f_{s}$, where $f_{s}$ is the signal sampling frequency. The micro-Doppler frequency $f_{md}$ is a cosine signal, so the sampling frequency should satisfy $f_{s}>2f_{md}$. A and Δ have a linear relationship, namely A=mΔ, where Δ is a discrete step. The relationship between ω and $f_{md}$ is complex, affected not only by the linear relationship, but also by nonlinear sinusoidal relationships. For simplicity, the equal interval discretization approximation can be used. ω is represented by the corresponding digital angular frequency Ω, that is, $\omega =\Omega f_{s}$. Because of $f_{s}>2f_{md}$, $\Omega =2\pi f_{md}/f_{s}<\pi$. Therefore, Ω can be discretized evenly in the range of [0,π).

If the amplitude $A_{\mu }$ and frequency $f_{\mu }$ of the micro-Doppler frequency are obtained by FFT, the vibration frequency of a vibration point and plane is $\omega_{v}=2\pi f_{\mu }$. The vibration amplitude of a vibration point is $A_{v}=\lambda A_{\mu }/2\omega_{v}$. The vibration amplitude of a vibration plane is $A_{v}=\lambda A_{\mu }/\omega_{v}$. The rotation velocity of the blade is $\Omega =2\pi f_{\mu }$ and the blade length is $L=\lambda A_{\mu }/\Omega$.

## 5. Computer Simulation


*Simulation experiment 1: Micro-Doppler effect of a typical single component*


Research on the micro-Doppler effect of the reflected waves of simple harmonic vibration points and planes and a rotation propeller by sparse reconstruction combined with time–frequency distributions can reveal the changing law of time–frequency and amplitude characteristics and extract the micro-Doppler frequency of the reflected waves. Suppose the emission signal frequency is 30 kHz, the signal length is 300 ms, and the sampling frequency is 300 kHz. When the sound velocity is 1500 m/s, the signal wavelength is 5 cm. The amplitude of the vibration point is 2 cm, and the vibration frequency is 100 Hz. In the case of a radius of the vibration circular plane of 1 m, vibration amplitude of 4 cm, vibration frequency of 60 Hz, blade length of a 5-blade propeller of 1 m, and rotational velocity of 10 Hz, the reflected waves, time–frequency distribution, and micro-Doppler frequency of three micro-motion components are shown in Figure 5, Figure 6 and Figure 7, respectively.

The micro-Doppler frequency of a vibration point varies with a sine law. The micro-Doppler effect of a vibration plane and propeller is more complex. Besides the micro-Doppler frequency presenting sinusoidal variation, the reflected wave amplitude also appears as a flash effect.

Compared with the vibration point and plane, blades have a larger size, so the micro-Doppler effect is more obvious. The five micro-Doppler frequencies of a 5-blade propeller are shown in Figure 7c. Since each blade with equal length is installed at equal intervals, the amplitude and frequency of the five micro-Doppler frequencies are the same, and the phase difference between blades is 2π/5.

Simulation experiment 2: Sparse representation analysis of multi-component superimposed wave

The vibration amplitude of the vibration point is set to 1 cm and the vibration frequency to 100 Hz. The vibration amplitude of the vibrating plane is set to 1 cm and the vibration frequency to 60 Hz, with a 5-blade propeller with a length of 1 m and a speed of 10 Hz. The experimental results are simulated as follows.

Let the target echo be the superposition of the reflected waves of a simple harmonic vibration point and plane and rotation blades. The echo is sparsely decomposed into the designed MicroDopplerlet time–frequency atomic dictionary by MP, and the time–frequency analysis is performed on the reflected wave of separated single components to extract the micro-Doppler frequency. The parameters are extracted, including the vibration amplitude and frequency of the point and plane, the length of the blade, and the rotational velocity.

In simulation experiment 2, the micro-motion parameters of the three components’ emission signal parameters are the same as those in simulation 1. The signal-to-noise ratio is 15 dB. The sparse decomposition grids are divided as follows: amplitude interval of micro-motion is 1 mm, frequency interval of micro-motion is 1 Hz, and time delay is 1 ms. The number of MP iterations is 10. The simulation results are shown in Figure 8, Figure 9, Figure 10 and Figure 11.

Figure 8 shows the MicroDopplerlet time–frequency atoms of the vibration point, plane, and blades. These time–frequency atoms match the respective modulation forms of the reflected waves. Figure 9 shows that the designed time–frequency atoms and MP are used to decompose and separate the echo of the three components correctly. Figure 10a shows the time–frequency distribution. In Figure 10b, the five sine waves with a large amplitude and low frequency are the micro-Doppler frequencies of the five blades. The micro-Doppler frequency amplitude can reach 1.25 kHz. The two sinusoidal waves with a smaller amplitude and higher frequency are the micro-Doppler frequencies of the point and plane, respectively. The micro-Doppler frequency amplitude of the vibration point is about 510 Hz, and that of the vibration plane is about 300 Hz. Figure 11 compares the true and estimated values of the micro-motion parameters of the point, plane, and blades. (a), (b), and (c) are the true values, and (d), (e), and (f) are the estimated values; the extracted parameters are consistent with the true values. The simulation results show that the micro-Doppler frequency processed by FFT can correctly estimate the parameters, such as the velocity and length of blades, the vibration amplitude, and the frequency of the vibration point and plane.

## 6. Conclusions

Based on the analysis of the micro-motions of an underwater target, the echo models of a simple harmonic vibration point and plane and rotation propeller were established in this paper, and the complex modulation characteristics of micro-Doppler effects are revealed. A method of sparse reconstruction combined with time–frequency distributions was presented, and a MicroDopplerlet time–frequency atomic dictionary matching with echo modulation was designed to realize the sparse decomposition of echoes and the analysis of micro-Doppler effects. The conclusions are as follows:(1)The vibration amplitude of the underwater target shell is at the centimeter level, and the vibration frequency ranges from tens of hertz to hundreds of hertz. The micro-Doppler frequency shift can reach tens of hertz to hundreds of hertz. The propeller has a large blade length and rotation rate, and its micro-Doppler frequency shift can reach hundreds to thousands of hertz;(2)The computer simulation shows that using the designed MicroDopplerlet time–frequency atomic dictionary not only separates vibration points, vibration surfaces, and propellers but can also distinguish the superimposed modulation components of the same type of components. In addition, the micro-Doppler frequency and vibration frequency, vibration amplitude, and the propeller rotation speed and length parameters of each micro-motion component are extracted. The extracted parameters are consistent with the given results, proving the correctness of the designed time–frequency atomic dictionary and the established model.

## Figures and Tables

**Figure 1 sensors-23-08066-f001:**
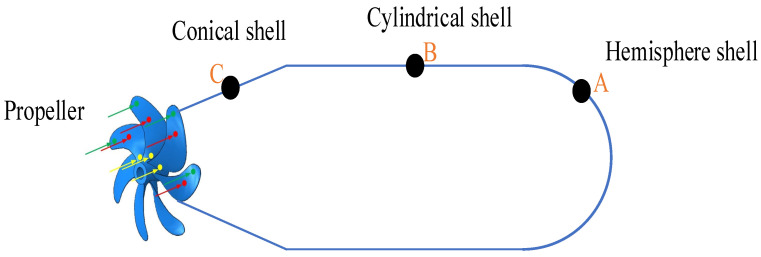
Simplified model of submarine.

**Figure 2 sensors-23-08066-f002:**
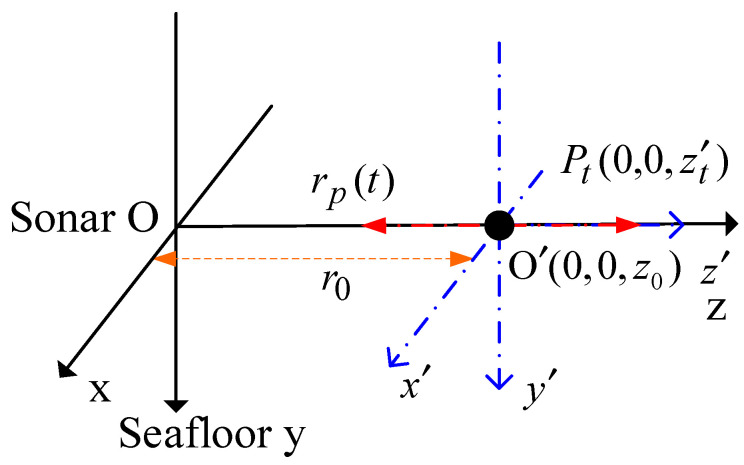
Geometric relationship between vibration point and sonar.

**Figure 3 sensors-23-08066-f003:**
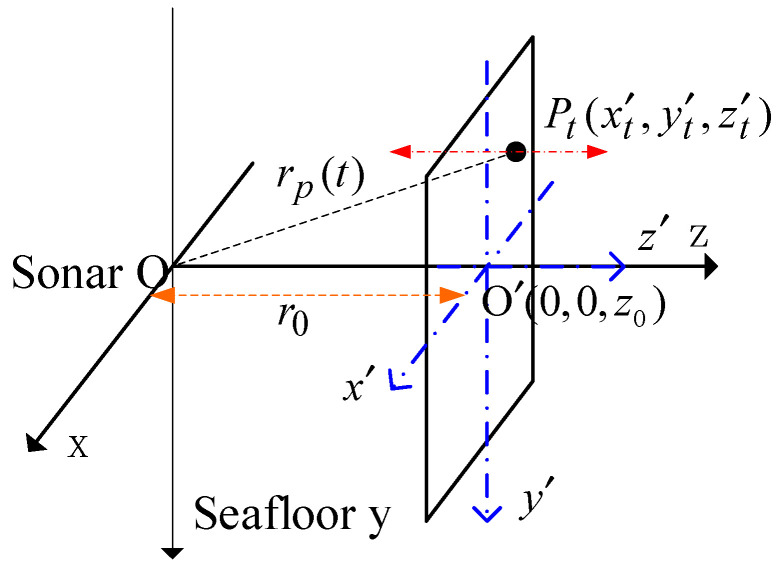
Geometric relationship between vibration plane and sonar.

**Figure 4 sensors-23-08066-f004:**
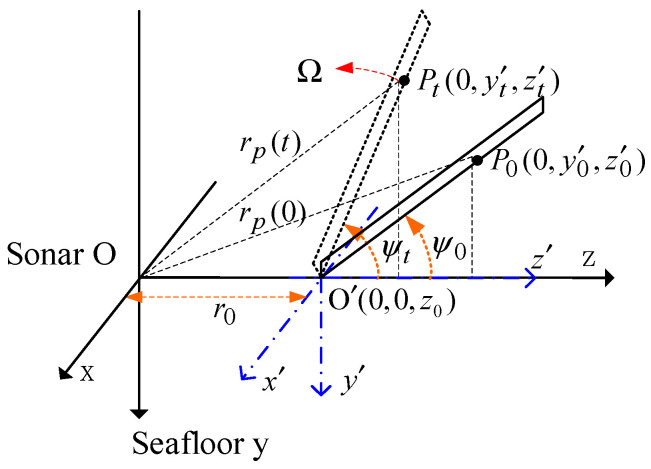
Geometric relationship between blades and sonar.

**Figure 5 sensors-23-08066-f005:**
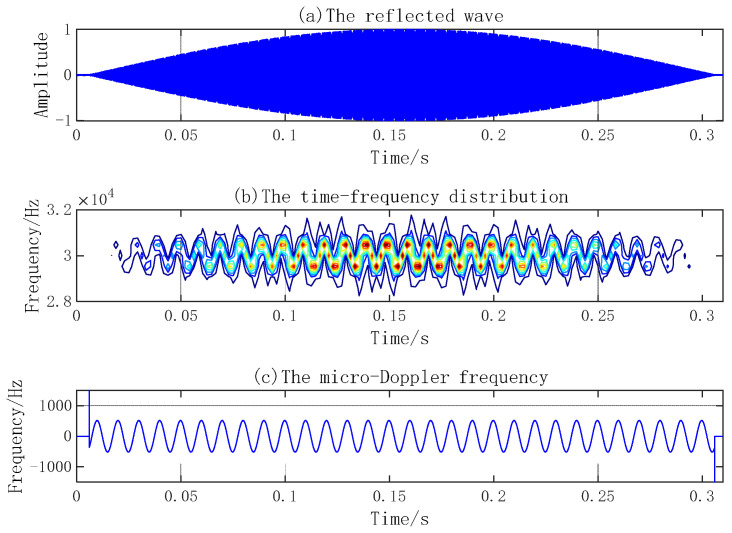
Micro-Doppler effect of vibration point.

**Figure 6 sensors-23-08066-f006:**
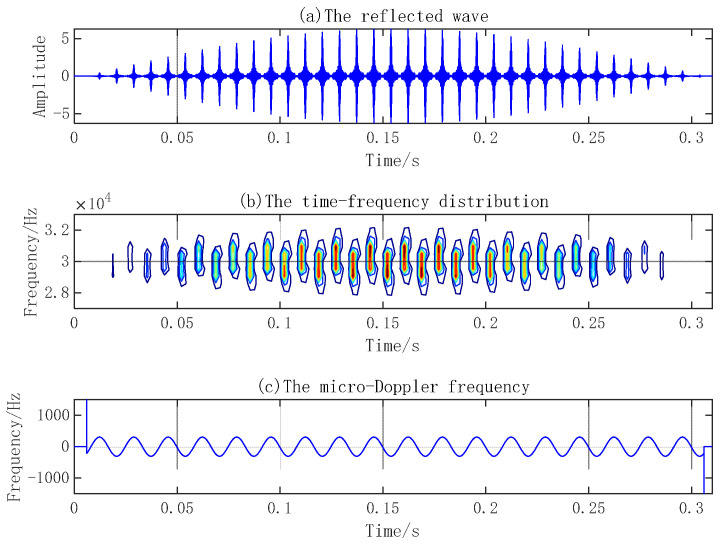
Micro-Doppler effect of vibration plane.

**Figure 7 sensors-23-08066-f007:**
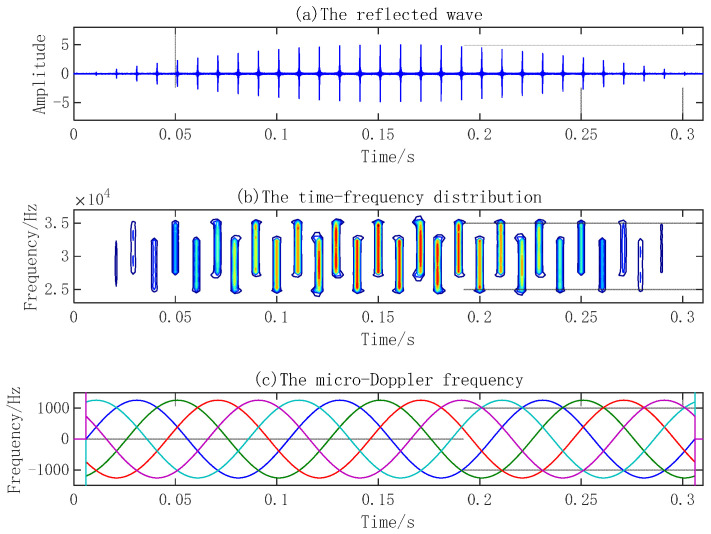
Micro-Doppler effect of propeller.

**Figure 8 sensors-23-08066-f008:**
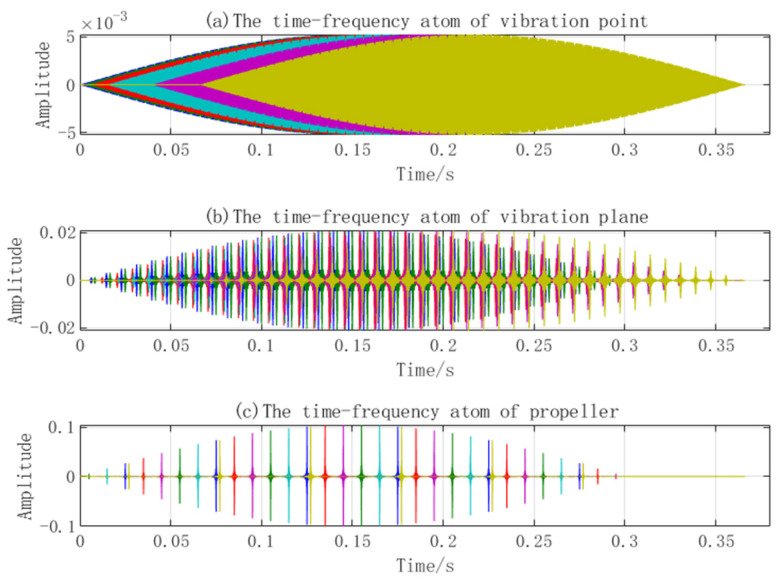
MicroDopplerlet time–frequency atoms.

**Figure 9 sensors-23-08066-f009:**
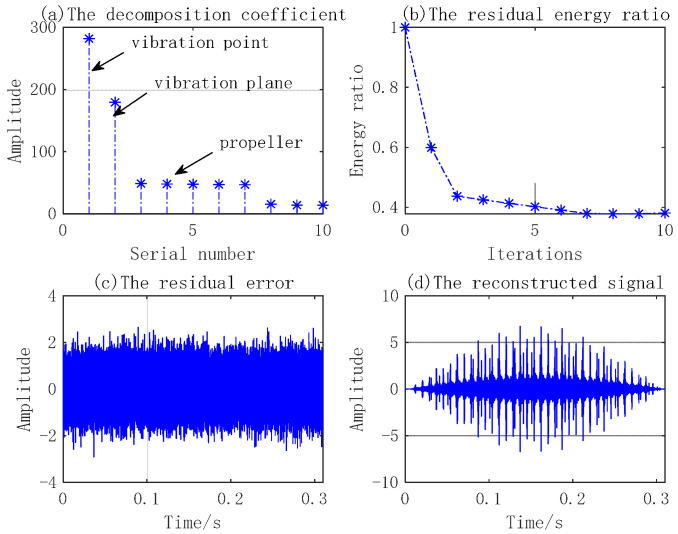
Sparse decomposition process.

**Figure 10 sensors-23-08066-f010:**
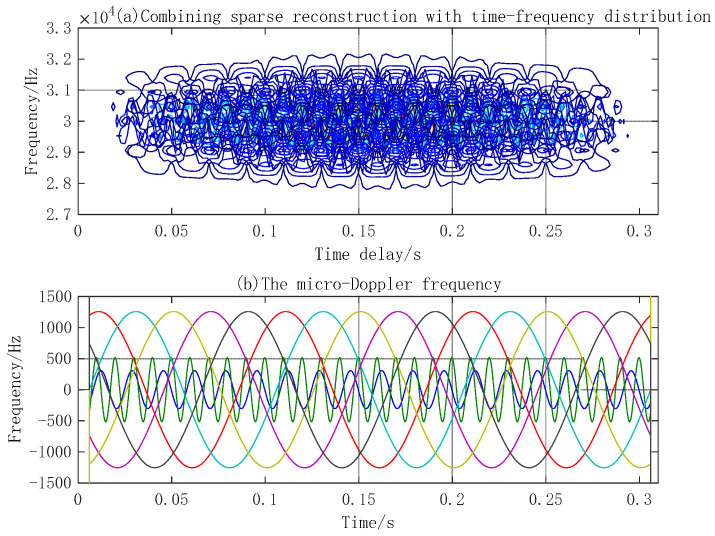
Time–frequency analysis.

**Figure 11 sensors-23-08066-f011:**
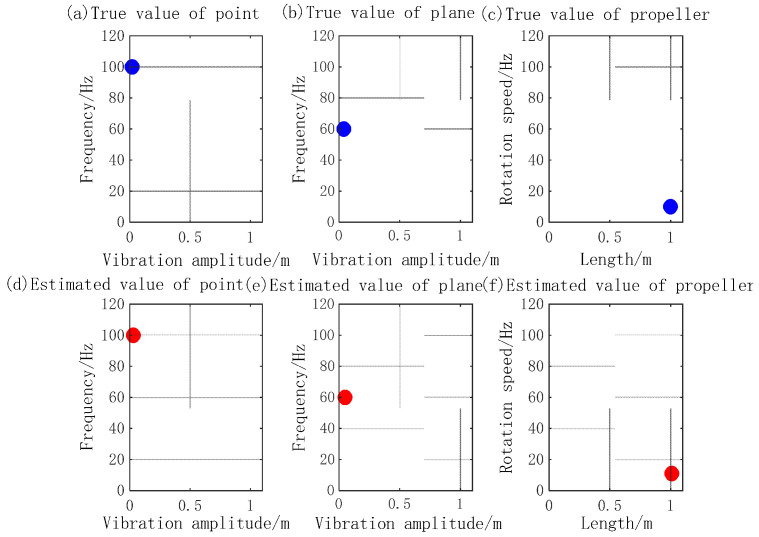
Micro-motion parameter estimation.

**Table 1 sensors-23-08066-t001:** Estimation of micro-motion amplitude, micro-motion frequency, and micro-Doppler frequency of propellers and different shells.

Category	Component	Main Parameters	Micro-Motion Amplitude	Micro-Motion Frequency	Micro-Doppler Frequency Shift
Propeller	5 blades	Blade length: 1.45 m, speed: 600 rpm	1.45 m	50 Hz	1.821 kHz
	7 blades	Blade length: 1.45 m, speed: 110 rpm	0.485 m	12.83 Hz	111.476 Hz
Single-layer shell	Conical shell C	Material: carbon steel, maximum diameter: 6.25 m, minimum diameter: 1 m, thickness: 0.03 m	7.995 cm	92 Hz	923.838 Hz
	Cylindrical shell B	Material: carbon steel, length: 45 m, thickness: 0.03 m, diameter: 6.5 m	5.657 cm	90 Hz	639.467 Hz
	Hemisphere shell A	Material: carbon steel, diameter: 6.5 m, thickness: 0.03 m	2.527 cm	95 Hz	301.522 Hz
Double-layer shell	Conical shell C	Material: steel, length: 4.87 m, skin thickness: 0.032 m, inner shell thickness: 0.04 m, minimum diameter: 5.98 m, maximum diameter: 7.5 m	1.003 cm	90 Hz	113.379 Hz
	Cylindrical shell B	Material: steel, length: 42 m, skin thickness: 0.032 m, inner shell thickness: 0.04 m, diameter: 7.5 m	2.521 cm	55 Hz	174.151 Hz
	Hemisphere shell A	Length: 7 m, skin thickness: 0.032 m, inner shell thickness: 0.04 m, diameter: 7.5 m	1.784 cm	92 Hz	206.145 Hz

## Data Availability

Not applicable.

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
