# Peer review of "Micro-Doppler Effect and Sparse Representation Analysis of Underwater Targets"

_sensors, 2023, doi:10.3390/s23198066_

Round 1

Reviewer 1 Report

This paper investigates the Micro-Doppler effect for underwater target detection and recognition. The work is of interest to general underwater acoustic researchers.

1. Please proofread the manuscript, especially the abstract and conclusion.

2. In Table 1, how to estimate the Micro-motion amplitude, frequency, and Doppler from the parameters? Are there theoretical formulas for calculation?

3. Have the author performed the experimental evaluation? How is the micro-Doppler in the realistic underwater environment?

4. The writing of conclusions is more like a discussion. Section 6 can discuss the findings in detail point by point. And a more general conclusion can be added.

The writing is fine. Please proofread.

Author Response

Dear experts:

Thank you for your suggestions for our paper. Here are our responses to experts’ opinions and suggestions.

1. Please proofread the manuscript, especially the abstract and conclusion.

Answer: We have carefully read the article and made revisions to the abstract and conclusion sections, with the modifications highlighted in red.

2. In Table 1, how to estimate the Micro-motion amplitude, frequency, and Doppler from the parameters? Are there theoretical formulas for calculation?

Answer: We have added in the paper.

3. Have the author performed the experimental evaluation? How is the micro-Doppler in the realistic underwater environment?

Answer: We conducted relevant experiments in an anechoic water tank and truly verified the existence of underwater micro-Doppler effect.

4. The writing of conclusions is more like a discussion. Section 6 can discuss the findings in detail point by point. And a more general conclusion can be added.

Answer: We have made modifications to the article and highlighted it in red.

Reviewer 2 Report

1.   The authors establish the echo models of harmonic vibration point and plane and rotating propeller based on the micro-motion analysis of underwater target. The complex modulation law of micro-Doppler effect was revealed. Aiming at the problem of the echo is a multi-component signal superposed by multiple modulated signals, and gives a method of sparse reconstruction combining with time-frequency distribution, and realizes signal separation and time-frequency analysis in this paper.

2.       Please elaborate the concept of the micro-Doppler effect in detail.

3.       Please compare the contributions of the proposed micro-motion analysis of underwater target to related technologies, in detail.

4.     The manuscript has 27 equations; the number of the equations should be decreased to retain the reader interest.

5.  In the figure 11, micro-motion parameter estimation, should be elaborated in detail.

Minor editing of English language required.

Author Response

Dear experts:

Thank you for your suggestions for our paper. Here are our responses to experts’ opinions and suggestions.

1. Please elaborate the concept of the micro-Doppler effect in detail.

Answer:  micro-Doppler effect:When, in addition to the constant Doppler frequency shift induced by the bulk motion of a radar target, the target or anystructure on the target undergoes micro-motion dynamics, such as mechanical vibrations or rotations, the micro-motion dynamicsinduce Doppler modulations on the returned signal, referred to as the micro-Doppler effect.

Due to the widespread application of micro-Doppler effect in radar, this concept is not elaborated in this article.

2. Please compare the contributions of the proposed micro-motion analysis of underwater target to related technologies, in detail.

Answer: There is relatively little researches on the micro-Doppler effect underwater, and there are few references and comparisons available. This article establishes moving model of micro motion components, these works are relatively cutting-edge in underwater environments and have certain research and application value. They can provide research ideas for subsequent underwater target recognition, true and false target identification, and other fields.

3. The manuscript has 27 equations; the number of the equations should be decreased to retain the reader interest.

Answer: There are indeed many formulas in the article, but deleting some may affect the coherence and completeness of reading. Therefore, we hope to remain unchanged.

4. In the figure 11, micro-motion parameter estimation, should be elaborated in detail.

Answer: We have added the relevant description in paper, and modified parts marked in red.

Round 2

Reviewer 1 Report

The authors have addressed the reviewer's comments.

Proofread and double-check the manuscript before publication.